# Association between Wine Consumption with Cardiovascular Disease and Cardiovascular Mortality: A Systematic Review and Meta-Analysis

**DOI:** 10.3390/nu15122785

**Published:** 2023-06-17

**Authors:** Maribel Lucerón-Lucas-Torres, Alicia Saz-Lara, Ana Díez-Fernández, Irene Martínez-García, Vicente Martínez-Vizcaíno, Iván Cavero-Redondo, Celia Álvarez-Bueno

**Affiliations:** 1Health and Social Research Center, Universidad de Castilla-La Mancha, 16071 Cuenca, Spain; mariaisabel.luceron@uclm.es (M.L.-L.-T.); ana.diez@uclm.es (A.D.-F.); irene.mgarcia@uclm.es (I.M.-G.); vicente.martinez@uclm.es (V.M.-V.); ivan.cavero@uclm.es (I.C.-R.); celia.alvarezbueno@uclm.es (C.Á.-B.); 2Facultad de Ciencias de la Salud, Universidad Autónoma de Chile, Talca 4810101, Chile; 3Universidad Politécnica y Artística del Paraguay, Asunción 2024, Paraguay

**Keywords:** cardiovascular mortality, cardiovascular disease, coronary heart disease, wine, adult people

## Abstract

**Background**: The objective of this systematic review and meta-analysis was: (i) to examine the association between wine consumption and cardiovascular mortality, cardiovascular disease (CVD), and coronary heart disease (CHD) and (ii) to analyse whether this association could be influenced by personal and study factors, including the participants’ mean age, the percentage of female subjects, follow-up time and percentage of current smokers. **Methods**: In order to conduct this systematic review and meta-analysis, we searched several databases for longitudinal studies from their inception to March 2023. This study was previously registered with PROSPERO (CRD42021293568). **Results**: This systematic review included 25 studies, of which the meta-analysis included 22 studies. The pooled RR for the association of wine consumption and the risk of CHD using the DerSimonian and Laird approach was 0.76 (95% CIs: 0.69, 0.84), for the risk of CVD was 0.83 (95% CIs: 0.70, 0.98), and for the risk of cardiovascular mortality was 0.73 (95% CIs: 0.59, 0.90). **Conclusions**: This research revealed that wine consumption has an inverse relationship to cardiovascular mortality, CVD, and CHD. Age, the proportion of women in the samples, and follow-up time did not influence this association. Interpreting these findings with prudence was necessary because increasing wine intake might be harmful to individuals who are vulnerable to alcohol because of age, medication, or their pathologies.

## 1. Introduction

The most prevalent cause of death globally is cardiovascular disease (CVD) [1], and the leading cause of life years lost is due to premature death [2]. Mortality from CVD is twice as high as mortality from other diseases, such as cancer, infectious diseases, or eating disorders [3]. Moreover, CVD is the primary cause of illness and morbidity [2], reducing patients’ quality of life [4]. Cerebrovascular diseases and coronary heart diseases (CHD) are two of the most prevalent CVDs [4].

A significant CVD risk factor is excessive alcohol drinking, which leads to 3 million deaths per year due to its harmful use and causes more than 200 diseases and disorders [5]. Alcohol consumption and cardiovascular events have a reported U or J-shaped association [6,7]. Certain amounts of alcohol consumption appear to have an adverse association with CHD due to their action on high-density lipoprotein cholesterol, preventing atherosclerosis [8]. Light to moderate alcohol use also lowers the prevalence of ischemic heart disease and improves the prognosis of people at risk of additional coronary events that could result in myocardial infarction [6]. A previous study reported that the dose of alcohol with the lowest health risk was between 0–7.5 drinks per week or 12.5 g per day, and the highest health risk was when consuming about 38 g of alcohol per day or the equivalent of 23 drinks per week [9]. Additionally, light drinking has been linked to an elevated risk of cardiovascular disease, whereas heavy drinking has been linked to an exponential rise in CVD [10]. The theory of alcohol consumption as a potential protective for some pathologies conflicts between disciplines. From a pharmaceutical point of view, alcohol consumption interacts with multiple drugs, such as diuretics, narcotics, and antidepressants [11], among others, as it may cause pharmacokinetic interactions by altering the metabolism of the alcohol and/or the drug [12]. When alcohol metabolism is decreased, it can lead to increased levels of alcohol in the blood, which can be caused by drugs used for ulcers and heartburn, such as histamine H2 receptors [12]. In the case of wine, caution must be taken with regard to the interaction of resveratrol with certain drugs, as evidence has shown that alcohol can modify their metabolism. These alterations cause drugs to reach the blood in smaller quantities, as in the case of nifedipine, and oral anticoagulants, where high doses of resveratrol in anticoagulated patients may increase the risk of haematomas and haemorrhages [13,14].

From a psychiatric point of view, there is also a conflict with alcohol consumption, as acute, high-dose alcohol consumption increases the risk of suicide [15]. Finally, from a cardiologist’s point of view, a protective effect of light moderate alcohol and wine consumption on cardiovascular health has been reported over the years [16]. In recent years, studies based on Mendelian randomisation approaches questioned this effect, consisting of analyses from a genetic approach where it was observed that a reduced risk of coronary heart disease among carriers of the alcohol dehydrogenase 1B (ADH1B) gene was evident when they drank less alcohol, concluding that reducing alcohol consumption is beneficial for cardiovascular health [17]. However, no study has been stratified by the type of alcoholic beverage [18].

It seems that all alcohol does not have the same effect. There is controversy in the existing evidence as to which beverage could be less harmful to CVD [19]; nevertheless, if stratifying by alcoholic beverage intake, a positive effect has been observed between wine and nonfatal CHD, whereas drinking beer is linked to a higher risk of a non-fatal stroke [20]. According to this evidence, several of the components in wine, including but not limited to water, carbohydrates, organic acids, minerals, alcohol, and aromatic substances, may be beneficial to health [21]. Some of these substances, when supplemented in the diet, have demonstrated positive effects on CVD [22] and cognitive impairment [23]. The positive effects of polyphenols, including resveratrol and phenolic acids, on CHD have also been reported [20]. Resveratrol is the most important polyphenol of the non-flavonoid family, along with the tannins [24]. Although there is previous evidence to question the benefits of resveratrol, on the one hand, studies have reported on the benefits of resveratrol consumption for cardiovascular health [25], while others have shown how resveratrol in high doses in certain populations increases clinical cardiovascular values, enhancing cardiovascular risk [26], and may cause endothelial cytotoxicity and apoptosis [27]. Despite these findings, the resveratrol present in red wine appears to have many health benefits, as it is anti-inflammatory, antioxidant, and antimutagenic in diseases such as cancer [28]; it has an anti-neuroinflammatory effect [29] and neuroprotective against toxins for cognitive impairment; it inhibits LDL oxidation, promotes endothelial relaxation, suppresses platelet aggregation and has anti-atherosclerotic functions (i.e., it provides a host of benefits for cardiovascular health) [8]. Despite this, excessive alcohol consumption increases the risk of pathologies such as CHD mortality, cancers such as oesophageal and oral cancer, and cerebrovascular death [30].

Given the high incidence of CVD and its possible association with wine consumption [31], a systematic review and meta-analysis are needed to elucidate the nature of this relationship. Therefore, the aims of this manuscript were (i) to examine the association between wine consumption and cardiovascular mortality, CVD, and CHD and (ii) to analyse whether this association could be affected by individual and study characteristics, including the participants’ mean age, the percentage of female subjects, follow-up time and the percentage of current smokers.

## 2. Methods

The Cochrane Collaboration Handbook [32] provided the guidelines for conducting this meta-analysis, which was reported in accordance with the MOOSE (Meta-analysis of Observational Studies in Epidemiology) statement [33]. The registration number for this study in PROSPERO is CRD42021293568.

### 2.1. Search Strategy

From their establishment until 26 March 2023, the PubMed, Scopus, and Web of Science databases were searched. Following the PICOS strategy (population, intervention/exposure, comparison, outcome, and study design), the following free terms were joined with Boolean operators to carry out a systematic search: “adults”, “young adults”, “elderly adults”, “older adults”, “adult population”, “adult subjects”, “alcohol”, “wine”, “alcohol consumption”, “wine consumption”, “CVD”, “cardiovascular disease”, “coronary heart disease”, “heart failure”, “cardiovascular events”, “coronary artery disease”, “myocardial infarction”, “cardiovascular outcomes”, “mortality”, “cardiovascular mortality”, “cardiovascular death”, “cohort”, “cases and controls”, “longitudinal studies”, and “prospective studies” (Appendix A). References from previous meta-analyses or systematic reviews were also evaluated.

### 2.2. Selection Criteria

The systematic review and meta-analysis included studies on the relationship between wine consumption and CVD, CHD, and cardiovascular mortality in order to compare the impact of wine on participants who drank it against those who did not. Cardiovascular mortality is defined as deaths from any cardiovascular event, including heart and blood vessel diseases collectively referred to as CVD, and when the arteries that carry oxygen to the heart muscle stiffen, it is known as CHD. The following were the inclusion criteria: (i) population: subjects who were older than 18 at the start; (ii) exposure: wine consumption; (iii) outcome: cardiovascular events such as CHD, CVD, and cardiovascular mortality; (iv) study design: cohort and case–control studies (longitudinal studies). We excluded studies when they (i) were review articles, ecological studies, editorials, or case reports; (ii) in languages other than English or Spanish; or (iii) did not differentiate wine use from other alcoholic drinks in its report.

### 2.3. Data Extraction and Risk of Bias Assessment

Table 1 summarises the major study characteristics, including details on (1) the references: date of publication and first author; (2) the country where the study’s data were compiled; (3) the design of the study (cohort or cases and controls studies); (4) population characteristics: sample size, gender representation, age, and population type (general population, men, women); (5) the follow-up of exposure (years); and (6) outcomes: cardiovascular events (cardiovascular mortality, CVD, and CHD).

The risk of bias in cohort studies was evaluated using the Quality Assessment Tool for Observational Cohort and Cross-Sectional Studies [34] from the National Heart, Lung, and Blood Institute of the United States based on the following areas: (1) research question, (2) study population, (3) participation rate, (4) recruitment, (5) sample size, (6) timeframe for associations, (7) exposure levels, (8) exposure measures, (9) assessment, (10) repeated exposure assessment, (11) outcome measures, (12) the blinding of exposure assessors, (13) loss to follow-up, and (14) statistical analyses.

**Table 1 nutrients-15-02785-t001:** Main characteristics of the included studies.

Reference	Country	Design of Study	Characteristics of the Participants			Follow Up (Years)	Outcome
			N, Women (%)	Age (SD)	Type of Population		
Kaufman et al. 1985 [35]	EE.UU.	Case–Control	Cases: 123 Controls: 66	Cases: 30 to 54 Controls: 30 to 54	Men	NR	CHD
Klatsky et al. 1986 [36]	EE.UU.	Cohort	13,056 (55.8)	-	General population	5	CHD
Klatsky et al. 1990 [37]	EE.UU.	Cohort	28,488	40.5	General population	8	CHD CVD
Klatsky et al. 1992 [38]	EE.UU.	Cohort	128,934 (55.9)	30–70	General population	8	Mortality
Klatsky et al. 1993 [39]	EE.UU.	Cohort	17,527 (60.6)	41.4	General population	8	Mortality
Gronbaek et al. 1995 [40]	Denmark	Cohort	3498 (54.5)	30–70	General population	11	Mortality
Renaud et al. 1999 [41]	France	Cohort	22,093 (0)	49.22 (5.74)	Men	12–18	Mortality CVD
Theobald et al. 2000 [42]	Sweden	Cohort	385	18–65	General population	22	Mortality
Gronbaek et al. 2000 [43]	Denmark	Cohort	12,846 (44)	20–98	General population	NR	CHD
Tavani et al. 2001 [44]	Italy	Case–Control	Cases: 507 (25.4) Controls:478 (37.9)	45–70	General population	NR	CHD
Mukamal et al. 2003 [45]	EE.UU.	Cohort	38,077 (0)	53.9	Male health professionals	12	CHD
Marques-Vidal et al. 2004 [46]	France	Cohort	- France: 7352 - Northern Ireland: 2398	54.9 (2.9)	General population	4	CHD
Dorn et al. 2007 [47]	EE.UU.	Case–Control	Cases: 33 Controls: 360	Cases: 56.1 (8.5) Controls: 53.7 (9.8)	Women	6	CHD
Burke et al. 2007 [48]	Australia	Cohort	514 (49.8)	15–88	General population	11.6	CVD
Schröder et al. 2007 [49]	Spain	Case–Control	Cases: 244 (0) Controls: 1270 (0)	Cases: 58.8 (10.9) Controls: 50.2 (13.5)	Men	NR	CHD
Suadicani et al. 2008 [50]	Denmark	Cohort	3022 (0)	40–59	Men	16	Mortality
Gémes et al. 2016 [51]	Norway	Cohort	58,827 (54)	49.1 (16.9)	General population	11.6	CHD
Britton et al. 2016 [52]	United Kingdom	Cohort	7010 (70.66)	56 (6)	General population	25	Mortality
Tverdal et al. 2017 [53]	Norway	Cohort	115,592 (63)	41	General population	6	CVD
Ricci et al. 2018 [23]	Europe	Cohort	17,594 (43)	35–70	General population	12.5	CHD
Song et al. 2018 [54]	EE.UU.	Cohort	156,728 (8)	65.3 (12.1)	Veterans	2.9	CHD
Panagiotakos et al. 2019 [55]	Greece	Cohort	3042 (50.2)	18–88	General population (The ATTICA study)	8.4	CVD
Schutte et al. 2020 [56]	United Kingdom	Cohort	446,439 (53.8)	56.4 (8.1)	General population	4	CHD
Maugeri et al. 2020 [57]	Czech Republic	Cohort	1773 (54.95)	46.75	General population	NR	CVD
Schutte et al. 2021 [22]	United Kingdom	Cohort	354,969 (60.4)	56.7 (8.3)	General population (Alcohol consumers and never drinkers)	6.9	CHD CVD

Results are reported as median ± SD or interquartile range. CHD: Coronary heart disease; CVD: Cardiovascular disease; NR: Not reported.

The risk of bias in the case–control studies were evaluated in accordance with the following domains using the Quality Assessment of Case-Control Studies [58] from the National Heart, Lung, and Blood Institute of the United States: (1) research question, (2) study population, (3) target population, (4) sample size, (5) recruitment, (6) inclusion and exclusion criteria, (7) case and control definitions, (8) random selection of study participants, (9) concurrent controls, (10) exposure measures and assessment, (11) blinding of exposure assessors, and (12) statistical analyses.

Finally, using both tools, the total risk of bias for each study was graded as “good” if the majority of the criteria were satisfied, “fair” if some criteria were met, or “poor” if there were few criteria met.

Two independent reviewers (M.L.-L.T. and C.A.-B.) selected the studies, extracted the data, and determined the risk of bias. Conflicts were settled by consensus-building or with the assistance of a third researcher (A.S.-L.).

### 2.4. Data Synthesis and Statistical Analysis

Some methodological details needed to be considered. The meta-analysis only included the study with the greater sample size, where two studies used the same population. The relative risk (RR) and odds ratios (OR) for the association between wine consumption and cardiovascular mortality, CVD, and CHD were included together in the meta-analysis [59]. Studies that reported hazard ratios (HR) were converted to RR using the formula below: RR = (1 − *e*HRln(1 − *r*))/*r* [59].

The association between wine consumption and cardiovascular events (cardiovascular mortality, CVD, and CHD) was calculated using pooled estimates of relative risk (RR) and their corresponding 95% confidence intervals (95% CIs) using the DerSimonian and Laird random effects [60] method. The I2 statistic, which varied between 0% and 100%, was utilised to investigate the inconsistency in accordance with the guidelines in the Cochrane Handbook [61]. The inconsistency was classified as not important (0–30%), moderate (30–50%), substantial (50–75%), or considerable (75–100%) based on the I2 values. The corresponding *p* values were also taken into account. The τ^2^ statistic was also used to assess heterogeneity; it was classified as low when τ^2^ was less than 0.04, moderate when τ^2^ was between 0.04 and 0.14, and substantial when τ^2^ was between 0.14 and 0.40 [62].

To determine the reliability of the summary estimates, sensitivity analysis was methodically carried out by removing the studies one at a time. Random effects meta-regression analyses were used to address whether participants’ mean age, the percentage of females, follow-up time, and the percentage of current smokers with wine exposure could modify the association of wine consumption and cardiovascular events (cardiovascular mortality, CVD, and CHD). Although the included studies reported analyses that included different confounding variables, we were only able to perform meta-regressions for the mean age, percentage of females, follow-up time, and percentage of current smokers because these were the variables most commonly reported by the studies included in our analyses. Egger’s regression asymmetry test [63], which was applied at a threshold of <0.10, was lastly employed to evaluate whether a publication bias was present.

STATA SE software, version 15 (StataCorp, College Station, TX, USA) was used for all statistical analyses.

## 3. Results

### 3.1. Systematic Review

The search retrieved 7042 articles. In total, 128 studies were chosen after duplicates were removed by reviewing the title and abstract. Finally, this systematic review comprised 25 studies [22,23,35,36,37,38,39,40,41,42,43,44,45,46,47,48,49,50,51,52,53,54,55,56,57], and the meta-analysis included 22 studies [22,23,35,36,38,39,40,41,42,43,44,45,46,47,48,49,50,51,52,53,54,55,57] (Figure 1). Four case–control studies [35,44,47,49] and 21 cohort studies [22,23,36,37,38,39,40,41,42,43,45,46,48,50,51,52,53,54,56,57] were among the included studies. The studies were conducted in nine countries: eight in the EEUU [35,36,37,38,39,45,47,54], three in Denmark [40,43,50] and the United Kingdom [22,52,56], two in Norway [51,53], and France [41,46], and one each in Sweden [42], Australia [48], Greece [55], Czech Republic [57], Italy [44] and Spain [49]. A total of 1,443,245 subjects (ages that ranged from 18.0 to 98.0 years) participated in studies that were conducted between 1985 and 2021. Data from the same sample were given by two studies [37,56], and the meta-analysis included the study with a higher sample size. The follow-up period ranged from 4 to 25 years. Regarding cardiovascular events, seven studies reported CVD [22,37,41,48,53,55,57], 14 studies reported CHD [22,23,35,36,37,43,44,45,46,47,49,51,54,56], and seven studies reported cardiovascular mortality [38,39,40,41,42,50,52]. The effect of wine on the quantity of wine consumed could not be analysed because it was not reported in many of the studies. The characteristics of the included studies are shown in Table 1. Finally, a different set of covariates was used to adjust the analyses reported by the included studies (Appendix A).

### 3.2. Risk of Bias Assessment

The overall risk of bias for the studies that were included and that examined the association between wine intake and CVD, CHD, and cardiovascular mortality was good in every study (100%).

All studies included information regarding the domains linked to exposure levels and the time frame to observe an effect for the risk of bias evaluation of cohort studies. Information on the blinding of the assessors was only mentioned in one study [40] (5%). In addition, 36.4% of the studies included sample size justification, power description, or variance and effect estimates, and there are just two studies [40,46] that evaluated the exposure domain on more than one occasion (10%) (Appendix A).

For the assessment of the risk of bias of case–control studies, two studies included information from all domains [47]. One study [35] did not provide information on the domain of case and control definitions (25%), and information about blinding exposure assessors was supplied by only one study [47] (25%) (Appendix A).

### 3.3. Meta-Analysis

DerSimonian and Lair random effect models were used to calculate the pooled RR for the effect of wine consumption on the risk of CHD, which was 0.76 (95% CIs: 0.69, 0.84); for the risk of CVD, it was 0.83 (95% CIs: 0.70, 0.98), and on the risk of cardiovascular mortality it was 0.73 (95% CIs: 0.59, 0.90). The heterogeneity of these estimates was substantial (τ^2^: 0.0185; τ^2^: 0.0226; and τ^2^: 0.0510; respectively) (Figure 2).

### 3.4. Sensitivity Analysis and Meta-Regression Models

When data from individual studies were removed from the analysis one at a time, the pooled RR estimations for the association of wine consumption with cardiovascular mortality, CVD, and CHD were not significantly changed (in magnitude or direction) (Appendix A).

The participants’ mean age, the percentage of women, the duration of follow-up, and whether or not the participants currently smoked were not associated with study heterogeneity according to the random-effects meta-regression models for the association between wine consumption and cardiovascular mortality, CVD, and CHD (Appendix A).

### 3.5. Publication Bias

Finally, Egger’s test showed publication bias evidence for the association between wine consumption and CVD (*p* = 0.003) but not for the association between wine consumption and CHD (*p* = 0.162) or cardiovascular mortality (*p* = 0.762).

## 4. Discussion

According to our data, there was an inverse association between the consumption of wine and the risk of CHD, CVD, and cardiovascular mortality. The participants’ mean age, the percentage of women, the duration of follow-up, or if the population currently smoked did not appear to have an effect on this association. Our meta-analysis added to the previous evidence on the relationship between wine consumption and different cardiovascular events, distinguishing cardiovascular mortality, CVD, and CHD.

The positive effects of wine on cardiovascular mortality were first reported in 1979, indicating that moderate wine consumption could exert a protective effect against different pathologies [64]. Although these effects could be due to the components of wine, there is controversy as to whether they could also be attributed to ethanol, as it has been described that some components of wine could lose their effect in the absence of ethanol [65]. Moreover, it appears that de-alcoholised wines conserve antioxidant effects [66] and could produce a protective effect against thrombosis [67].

The main cause of death on a global scale is CVD. In 2017, CVD deaths increased to 17.79 million [68], in which ischemic heart disease was responsible for almost half of all cases of CVD [69]. Most CVDs can be prevented by acting on risk factors, early detection, and early treatment. A J-shaped relationship has been identified between wine consumption and cardiovascular events, according to previous meta-analyses [31,70], suggesting that low-to-moderate wine intake may help promote better health. Our results confirm the existing data that moderate wine consumption can be inversely associated with as cardiovascular outcomes such cardiovascular mortality, CVD, and CHD.

There are data supporting the effect of other forms of alcohol on CVD. In the case of beer, the effects differ according to the amount, the distribution of consumption, and the foods consumed during alcohol consumption, although moderate consumption is considered to minimise the risk of CVD and death from all causes [71]. Concerning spirits, consumption is often more isolated, limited to a few days of the week, especially on weekends, but can be consumed excessively on a single occasion [72,73]. This excessive and sporadic consumption could explain why there are no positive effects for CVD and explains why it increases the risk of developing certain diseases, such as ischemic heart disease [74]. Not only is the type of alcoholic beverage important for health, but also the dose of alcohol consumed. Previous evidence has shown that light to moderate drinking reduces the risk of all-cause mortality and mortality from diabetes or nephritis, but it should be noted that heavy drinkers have a higher risk of all-cause mortality and accidents as well as a significantly increased risk of developing and dying from cancer [65].

Wine seems to have a stronger beneficial effect on CVD than other alcoholic beverages, according to the evidence currently available [75]. The inverse association of wine consumption has been described for red and white wine; the differences in the strength of this association could be due to the different concentrations of some components [66]. The higher consumption of red wine [46] or the composition of grape skins, which, depending on the grape variety, are made up of different kinds of polyphenols, including gallic acid and polymeric anthocyanins, quercetin, myricetin, catechin, and epicatechin (flavonols), are what give the wine its antioxidant properties [76]. Additionally, a few explanations of the inverse relationship between wine and cardiovascular disease have been put out, including but not limited to (i) the phenolic compounds in wine, which reduce LDL oxidation, thrombosis risk, plasma and lipid peroxide, and (ii) its alcohol components, which reduce thrombosis risk and fibrinogen levels and induce collagen and platelet aggregation [77].

Our systematic review and meta-analysis have some limitations that should be mentioned. First, the amount of wine consumed was not reported specifically in any of the studies since numerous factors affected how they measured it. Second, by applying Egger’s test, we discovered evidence of publication bias. Third, we excluded the grey literature from the search and restricted it to research in English and Spanish. Fourth, most studies did not include information on assessor blinding or whether exposure was evaluated more than once over time after determining the risk of bias. Fifth, even though the WHO recommends using alcohol consumption limits to solve this issue, there is not a global consensus on safe drinking limits or a maximum suggested intake. Sixth, wine consumption among participants is likely to be combined with other types of alcohol that could interfere with the results. Seventh, we must consider as a limitation that not all studies assessed the same confounding variables such as age, sex, race, body mass index, smoking, marital status, education, and some pathologies such as diabetes; therefore, we could only perform meta-regressions for the mean age, percentage of women, follow-up time, and percentage of current smokers. Eighth, one of the limitations of our study is related to the reported amount of wine consumption, which differed between studies and was often not reported; therefore, we could not analyse the effect of the amount of wine consumed on cardiovascular health. Ninth, although some of the included studies reflected that the participants at baseline were healthy adults, others did not; therefore, although they were assumed to come from the general population with a general incidence of pathologies, it cannot be discarded that these studies could have included participants who stopped drinking because of health problems. Finally, due to a lack of evidence, we were unable to examine the relationship between wine intake and cardiovascular mortality, CVD, and CHD by wine type or sex. Although the analysis of the effect of wine consumption on total mortality could be of interest to our purpose, a lack of data prevented us from carrying out this analysis.

In conclusion, this systematic review and meta-analysis revealed an inverse association between wine consumption and CVD, CHD, and cardiovascular mortality. Age, the percentage of women in the samples, and follow-up time do not appear to have any effect on this association. Our findings should be interpreted with caution; thus, increasing wine consumption could be detrimental for patients who are vulnerable to alcohol due to age, medication, or pathology. There are current dietary recommendations that include wine consumption, as in the case of the Mediterranean diet. Given the findings of this meta-analysis, it would be interesting to suggest drinking wine as part of other dietary recommendations. Future research is needed to differentiate these effects by the type of wine.

## Figures and Tables

**Figure 1 nutrients-15-02785-f001:**
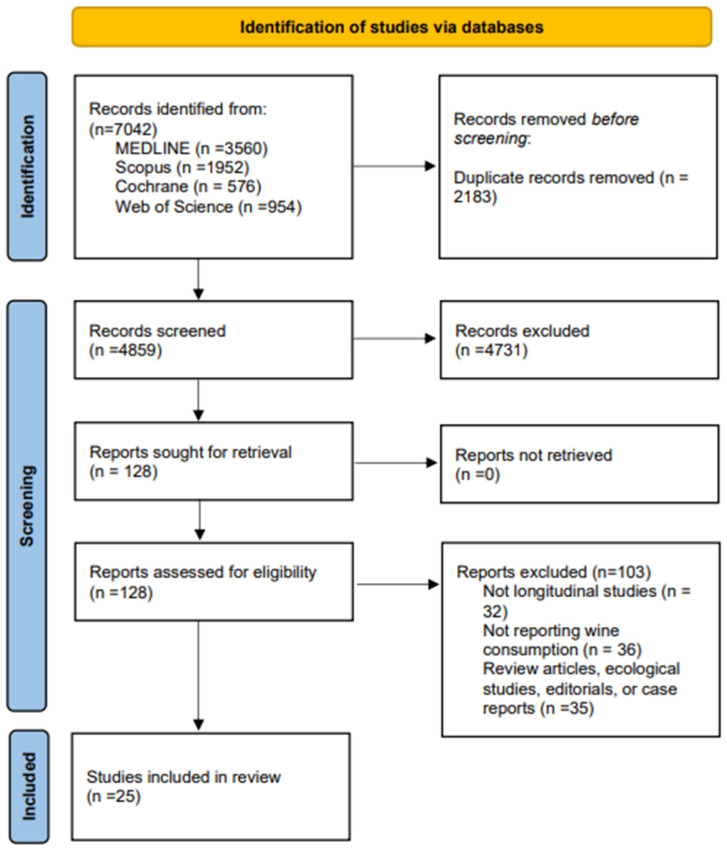
PRISMA 2020 flow diagram for new systematic reviews which included searches of databases.

**Figure 2 nutrients-15-02785-f002:**
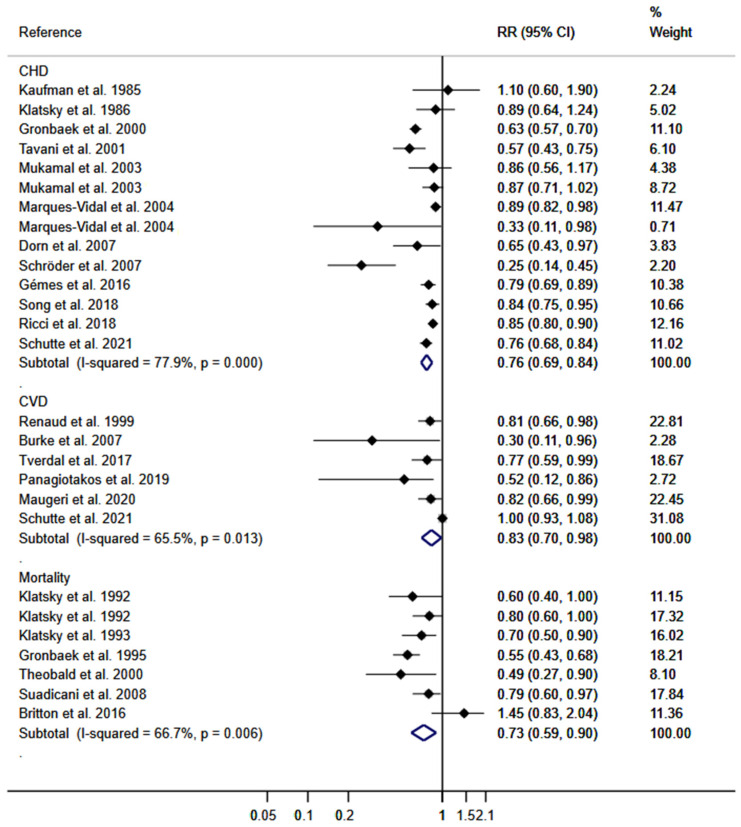
Meta-analysis for the association of wine consumption with CVD, CHD, and mortality for CV events. Horizontal lines represent the 95% confidence intervals of the study, and the black boxes represent the effect size of each study [22,23,35,36,38,39,40,41,42,43,44,45,46,47,48,49,50,51,52,53,54,55,57].

## Data Availability

Contact the authors by email.

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
