# Peer review of "Association between Wine Consumption with Cardiovascular Disease and Cardiovascular Mortality: A Systematic Review and Meta-Analysis"

_nutrients, 2023, doi:10.3390/nu15122785_

Round 1

Reviewer 1 Report

In the presented systematic review and meta-analysis, the authors tried to present the role of alcohol consumption on cardiovascular risk. Study properly designed, performed, presented and well discussed.
Alcohol consumption as an element of prevention is not easy, the problem is viewed differently by a clinical pharmacologist (risk of interaction, e.g. resveratrol - DOAC), a cardiologist (he sees mainly positive aspects), a psychiatrist (risk of addiction). These aspects must be emphasized, preferably in the introduction. Please also address the fact resvratrol vs polyphenols. According to some authors, resveratrol and cardiovascular system -- the unfulfilled hopes... Please also comment on the issue - do the consumption pattern and dose make the difference?

Author Response

Point-by-point response to reviewers’ comments (Manuscript ID: 2462588)

Reviewer(s)' Comments to Author:

Reviewer: 1

In the presented systematic review and meta-analysis, the authors tried to present the role of alcohol consumption on cardiovascular risk. Study properly designed, performed, presented and well discussed.

Comments to the Author:

Alcohol consumption as an element of prevention is not easy, the problem is viewed differently by a clinical pharmacologist (risk of interaction, e.g. resveratrol - DOAC), a cardiologist (he sees mainly positive aspects), a psychiatrist (risk of addiction). These aspects must be emphasized, preferably in the introduction. Please also address the fact resveratrol vs polyphenols. According to some authors, resveratrol and cardiovascular system -- the unfulfilled hopes... Please also comment on the issue - do the consumption pattern and dose make the difference?

Authors: We appreciate the reviewer’s comment. We have modified our manuscript with the suggestions in order to improve the quality of the manuscript.

INTRODUCTION

“The theory of alcohol consumption as a potential protective for some pathologies conflicts between disciplines. From a pharmaceutical point of view, alcohol consumption interacts with multiple drugs such as diuretics, narcotics, and antidepressants,11 among others, as it may cause pharmacokinetic interactions by altering the metabolism of alcohol and/or the drug.12 When alcohol metabolism is decreased, it can lead to increased levels of alcohol in the blood, which can be caused by drugs used for ulcers and heartburn, such as histamine H2 receptors.12 In the case of wine, caution must be taken with regard to the interaction of resveratrol and certain drugs, as evidence has shown that alcohol can modify their metabolism. These alterations cause drugs to reach the blood in smaller quantities, as in the case of nifedipine and oral anticoagulants, where high doses of resveratrol in anticoagulated patients may increase the risk of haematomas and haemorrhages.13,14

From a psychiatric point of view, there is also a conflict with alcohol consumption, as acute, high-dose alcohol consumption increases the risk of suicide.15 Finally, from the cardiologists' point of view, a protective effect of light moderate alcohol and wine consumption on cardiovascular health has been reported over the years.16 In recent years studies based on Mendelian randomisation approaches have questioned this effect, consisting of analyses from a genetic approach where it was observed a reduced risk of coronary heart disease among carriers of the alcohol dehydrogenase 1B (ADH1B) gene when they drank less alcohol, concluding that reducing alcohol consumption is beneficial for cardiovascular health.17 However, no study has stratified by type of alcoholic beverage.18

“Resveratrol is the most important polyphenol of the non-flavonoid family, along with the tannins.24 Although there is previous evidence to question the benefits of resveratrol, on the one hand, studies report benefits of resveratrol consumption for cardiovascular health,25 others have shown that resveratrol in high doses in certain populations increases clinical cardio-vascular values, increasing cardiovascular risk,26 and may cause endothelial cytotoxicity and apoptosis.27 Despite these findings, the resveratrol present in red wine appears to have many health benefits, as it is anti-inflammatory, antioxidant and antimutagenic in diseases such as cancer;28 has an anti-neuroinflammatory effect29 and neuroprotective against toxins X for cognitive impairment; inhibits LDL oxidation, promotes endothelial relaxation, suppresses platelet aggregation and has anti-atherosclerotic functions (i.e. it provides a host of benefits for cardiovascular health).8

“A previous study reported that the dose of alcohol with the lowest health risk was between 0-7.5 drinks per week or 12.5 grams per day and with the highest health risk when consuming about 38 grams of alcohol per day or the equivalent of 23 drinks per week.9

Reviewer 2 Report

In this manuscript entitled “ Association between wine consumption with cardiovascular disease and cardiovascular mortality: a systematic review and meta-analysis“ results are presented from a systematic review and meta-analysis of studies on the association between wine consumption and CVD. 

The manuscript reads well and the results are well presented.

Comments:

-        Defining wine consumption was very different between the studies that were selected; pooling all these different definitions into a simple ‘ yes’ or ‘no’ concept is very superficial and reduces the external validity of the study. Pooling occasional consumption of one glass of wine with daily consumption of a full bottle of wine is a mix  that limits the interpretation of the results

-        Alcohol consumption has been associated with numerous other health problems; the consumption of even one glass per day seems to be associated with a greater risk of certain cancers. Adding ‘ total mortality’ to the outcomes in this systematic review and meta-analysis would greatly enhance its interest.

-        If wine consumption is protective of CVD its incremental value should be examined in multivariate analysis; this was done in some of the studies but only age, sex, smoking and follow-up time were considered in this review. 

-        The subgroup of ‘ non wine consumers’ may include patients who stopped wine consumption because of health problems that may have increased their CVD risk. What are the results if the incidence of CVD during the first years after the survey were left out? 

-        In the 2021 ESC Guidelines on CVD prevention in clinical practice it is mentioned that “ Mendelian randomization studies do not support the apparently protective effects of moderate amounts vs. no alcohol against ASCVD, suggesting that the lowest risks for CVD outcomes are in abstainers and that any amount of alcohol uniformly increases BP and BMI” This is mainly based on a publication in BMJ 2014; 349, g4164 by Holmes et al. Does this conflicts with the results presented in this systematic review?

Author Response

Reviewer: 2

In this manuscript entitled “Association between wine consumption with cardiovascular disease and cardiovascular mortality: a systematic review and meta-analysis” results are presented from a systematic review and meta-analysis of studies on the association between wine consumption and CVD. 

The manuscript reads well and the results are well presented.

Comments to the Author:

- Defining wine consumption was very different between the studies that were selected; pooling all these different definitions into a simple ‘yes’ or ‘no’ concept is very superficial and reduces the external validity of the study. Pooling occasional consumption of one glass of wine with daily consumption of a full bottle of wine is a mix that limits the interpretation of the results.

Authors: We really appreciate the reviewer’s comment. We have considered the suggestion and added a sentence about this limitation in results and added a limitation commenting on this aspect.

“The effect of wine by quantity of wine consumed could not be analysed because it was not reported in many of the studies.

“Eighth, one of the limitations of our study is related to the reported amount of wine consumption between studies, which differs between studies and was often not reported, so we could not analyse the effect of the amount of wine consumed on cardiovascular health.”

Comments to the Author:

-        Alcohol consumption has been associated with numerous other health problems; the consumption of even one glass per day seems to be associated with a greater risk of certain cancers. Adding ‘total mortality’ to the outcomes in this systematic review and meta-analysis would greatly enhance its interest.

Authors: We are grateful for the reviewer's comment. We have commented on the suggestion made in the discussion of our study. In addition, a limitation in this regard has been added.

“Not only the type of alcoholic beverage is important for health, but also the dose of alcohol consumed. Previous evidence has shown that light to moderate drinking reduces the risk of all-cause mortality, and mortality from diabetes or nephritis, but it should be noted that heavy drinkers have a higher risk of all-cause mortality and accidents and had a significantly increased risk of developing and dying from cancer.75

“Although the analysis of the effect of wine consumption on total mortality could be of interest for our purpose, the lack of data prevents us of carrying out this analysis.”

Comments to the Author:

-        If wine consumption is protective of CVD its incremental value should be examined in multivariate analysis; this was done in some of the studies but only age, sex, smoking, and follow-up time were considered in this review. 

Authors: Thank you for the reviewer’s comment. We have addressed this suggestion in the methodology and discussion of our work.

“Although the included studies reported analyses including different confounding variables, we were only able to perform meta-regressions for mean age, percentage of females, follow-up time, and percentage of current smokers because these were the variables most commonly reported by the studies included in our analyses.

“Seventh, we must consider as a limitation that not all studies assessed the same confounding variables such as age, sex, race, body mass index, smoking, marital status, education, and some pathologies such as diabetes, so we could only perform meta-regressions for mean age, percentage of women, follow-up time, and percentage of current smokers.”

Comments to the Author:

-        The subgroup of ‘non wine consumers’ may include patients who stopped wine consumption because of health problems that may have increased their CVD risk. What are the results if the incidence of CVD during the first years after the survey were left out? 

Authors: Thanks to the reviewer for this appreciation. We have added a limitation on this aspect to improve the quality of our manuscript.

“Ninth, although some of the included studies reflected that participants at baseline were healthy adults, others do not, so although they were assumed to come from general population with a general incidence of pathologies, it cannot be discarded that could include participants who stopped drinking because of health problems.”

Comments to the Author:

-        In the 2021 ESC Guidelines on CVD prevention in clinical practice it is mentioned that “Mendelian randomization studies do not support the apparently protective effects of moderate amounts vs. no alcohol against ASCVD, suggesting that the lowest risks for CVD outcomes are in abstainers and that any amount of alcohol uniformly increases BP and BMI” This is mainly based on a publication in BMJ 2014; 349, g4164 by Holmes et al. Does this conflicts with the results presented in this systematic review?

Authors: We thank the reviewer for his suggestion. We have modified our manuscript to include the evidence from the cited study, thus enhancing the quality of our work.

“Finally, from the cardiologists' point of view, a protective effect of light moderate alcohol and wine consumption on cardiovascular health has been reported over the years.16 In recent years studies based on Mendelian randomisation approaches have questioned this effect, consisting of analyses from a genetic approach where it was observed a reduced risk of coronary heart disease among carriers of the alcohol dehydrogenase 1B (ADH1B) gene when they drank less alcohol, concluding that reducing alcohol consumption is beneficial for cardiovascular health.17 However, no study has stratified by type of alcoholic beverage.18

Round 2

Reviewer 1 Report

The authors made significant changes to the manuscript, so in my opinion it may be considered for publication